# *Vegfr3*-tdTomato, a reporter mouse for microscopic visualization of lymphatic vessel by multiple modalities

Esther Redder[1,2☯], Nils Kirschnick[1,2☯], Stefanie Bobe[1,2], René Hägerling[2¤], Nils Rouven Hansmeier[2¤], Friedemann Kiefer[1,2]*

**1** European Institute of Molecular Imaging, University of Münster, Münster, Germany, **2** Max Planck Institute for Molecular Biomedicine, Münster, Germany

☯ These authors contributed equally to this work.
¤ Current address: Institute of Medical Genetics and Human Genetics, Chartié – Universitätsmedizin Berlin, Berlin, Germany
* fkiefer@uni-muenster.de

**Data Availability Statement:** All raw data files are are available from the figshare database (DOI: https://doi.org/10.6084/m9.figshare.14931708.v1).

## Abstract

Lymphatic vessels are indispensable for tissue fluid homeostasis, transport of solutes and dietary lipids and immune cell trafficking. In contrast to blood vessels, which are easily visible by their erythrocyte cargo, lymphatic vessels are not readily detected in the tissue context. Their invisibility interferes with the analysis of the three-dimensional lymph vessel structure in large tissue volumes and hampers dynamic intravital studies on lymphatic function and pathofunction. An approach to overcome these limitations are mouse models, which express transgenic fluorescent proteins under the control of tissue-specific promotor elements. We introduce here the BAC-transgenic mouse reporter strain *Vegfr3*-tdTomato that expresses a membrane-tagged version of tdTomato under control of Flt4 regulatory elements. *Vegfr3*-tdTomato mice inherited the reporter in a mendelian fashion and showed selective and stable fluorescence in the lymphatic vessels of multiple organs tested, including lung, kidney, heart, diaphragm, intestine, mesentery, liver and dermis. In this model, tdTomato expression was sufficient for direct visualisation of lymphatic vessels by epifluorescence microscopy. Furthermore, lymph vessels were readily visualized using a number of microscopic modalities including confocal laser scanning, light sheet fluorescence and two-photon microscopy. Due to the early onset of VEGFR-3 expression in venous embryonic vessels and the short maturation time of tdTomato, this reporter offers an interesting alternative to *Prox1*-promoter driven lymphatic reporter mice for instance to study the developmental differentiation of venous to lymphatic endothelial cells.

## Introduction

The lymphatic vessel system consists of a blind-ending network of capillaries (capLV) that take up interstitial fluid, solutes, dietary lipids and cells, which together constitute the lymph. Subsequently, lymph is routed via collecting lymphatic vessels (colLV) to connections with

**Funding:** FK and ER are funded by Deutsche Forschungsgemeinschaft (DFG, German Research Foundation) - SFB1348/1 – 386797833 (https://gepris.dfg.de/gepris/projekt/386797833) and FK by Deutsche Forschungsgemeinschaft (DFG, German Research Foundation) - SFB1450/1 - 431460824 (https://gepris.dfg.de/gepris/projekt/431460824). FK and RH are funded by Deutsche Forschungsgemeinschaft (DFG, German Research Foundation) - SFB656/3 – 12467772 (https://gepris.dfg.de/gepris/projekt/12467772). NK is funded by CiM-IMPRS, the joint graduate school of the Cells-in-Motion Cluster of Excellence (EXC 1003 - CiM), University of Münster, Germany and the International Max Planck Research School - Molecular Biomedicine, Münster, Germany (https://gepris.dfg.de/gepris/projekt/194347757). The funders had no role in study design, data collection and analysis, decision to publish, or preparation of the manuscript.

**Competing interests:** The authors have declared that no competing interests exist.

subclavian veins where it enters the blood stream. Oak-leaf shaped lymphatic endothelial cells (LECs), with specialized junctions that form discontinuous, "button-like" cell-cell contacts facilitate fluid uptake into capLVs [1]. Transport of lymph within the fluid-tight colLVs is facilitated by intraluminal valves and coordinated contraction of smooth muscle cells [1–5]. Besides its essential function for the maintenance of tissue homeostasis, the lymphatic system is also indispensable for immune cell trafficking and dietary fat absorption, which also ensures the uptake of fat-soluble vitamins from the intestine [3, 6]. During embryonic development the first LECs differentiate within the cardinal and superficial veins [7, 8], while non-venous origins of LECs have been identified to contribute to lymph vessels (LV) in multiple vascular beds [9–11]. Development and maintenance of LVs are dependent on the key regulator VEGF-C, which signals via its receptor VEGFR-3 [10, 12–19]. The expression of VEGFR-3 in the mouse starts early during embryonic development in developing blood vessels but becomes largely restricted to the lymphatic vessels after midgestation [20]. Both, homozygous loss of either ligand or receptor result in prenatal death, deficient lymphatic sprouting and general vascular defects [14, 21].

In recent years, many factors have been identified that contribute to and are essential for LV development and maintenance during adulthood. Fluorescence-based visualization of vascular networks has profoundly contributed to this advancement. So far, different transgenic mouse lines have been described, which are suitable for fluorescent detection of LVs. Several lines are based on the expression of fluorescent proteins, e.g. GFP [22–24], mOrange2 [25] or tdTomato [26–28], under the control of the *Prox1* promoter. Furthermore, transgenic models using *Vegfr3* transcriptional elements to express fluorescent reporters are also reported [29–32]. However, the investigation and dynamic visualization of complex lymph vessel beds in specific organs are still underrepresented. A reason for this is the requirement for lymphatic reporters that encode bright fluorescent proteins as well as limited availability of wholemount immunostaining protocols with sufficient penetration depth, suited for the visualization of the LV beds in large volumes. Furthermore, access to complex imaging and tissue clearing techniques that preserve reporter protein fluorescence and allow for deep, three-dimensional (3D) volume imaging are still limited to a relatively small number of laboratories.

In response to this obvious demand, we decided to generate a novel lymphatic reporter mouse model that is suitable for multiple imaging modalities including confocal laser scanning (CLSM), light sheet fluorescence (LSFM) and two-photon laser scanning (2P-LSM) microscopy. This wide spectrum of microscopic modalities allows high resolution as well as dynamic or 3D visualization of the lymphatic network in large tissue volumes. Based on its relative brightness, excitation and emission spectra and fast maturation time, we expressed the tandem-dimer derivate of RFP, tdTomato [33] under transcriptional control of the *Vegfr3* promotor to label LECs in mice. Here, we show efficient labelling of LVs by the *Vegfr3*-tdTomato transgene in various murine tissues and demonstrate co-localization with immunostained VEGFR-3 and various identifying lymphatic markers. This confirms LEC-specific expression and makes this mouse line a valuable tool for LV imaging, as well as LEC FACS isolation. Moreover, the reporter can be useful for dynamic intravital microscopy of lymphatics. Tissue clearing and co-staining techniques in combination with LSFM allow for visualization of complex three-dimensional structures in so far under explored tissues.

## Results

### Generation of a *Vegfr3*-tdTomato transgenic mouse line

The VEGF-C / VEGFR-3 pathway is known to exhibit strong dose dependency. We therefore generated a transgenic model, in which a tdTomato-CAAX-pA cDNA was inserted into BAC

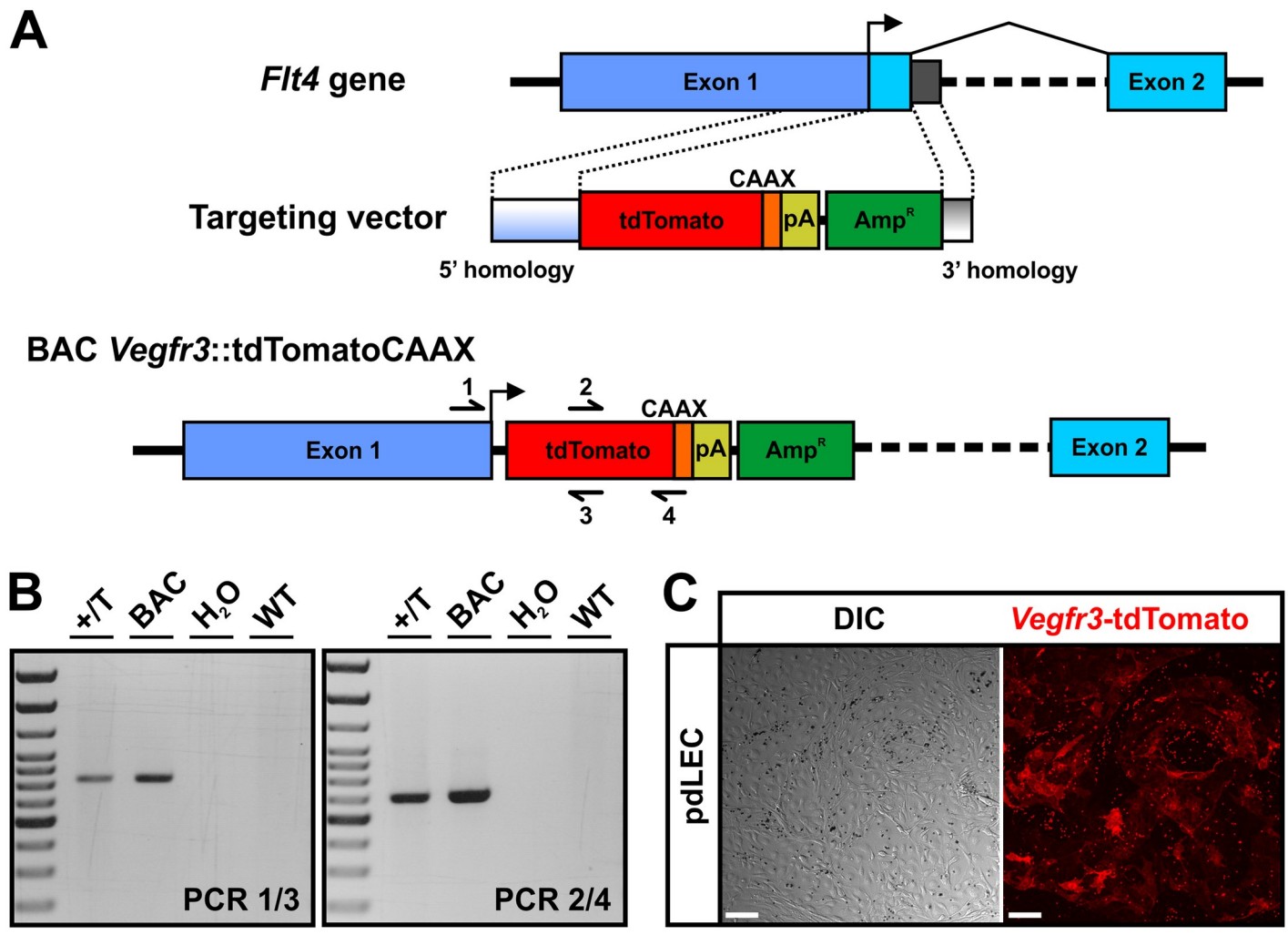

**Fig 1. Generation of the *Vegfr3*-tdTomato reporter mouse. A)** Schematic representation of the modified genomic region around the initiating ATG of the mouse *Flt4* gene in the BAC *Vegfr3*-tdTomato used for pronuclear injection. A cDNA expression cassette encoding the fluorescent protein tdTomato C-terminally fused to a CAAX-box for membrane-anchoring followed by a SV40-polyadenylation signal (pA) for positive selection was inserted into the BAC clone RP23-58E13 such that the initiating ATG of the *Vegfr3* was now usurped by tdTomato. For Red/ET recombineering, 3' and 5' *Vegfr3* homology regions derived from the genomic sequences preceding the initiating ATG and following Exon1 were added on both ends of the targeting cassette. Primers for PCR identification of recombinant clones and transgenic animals are indicated. **B)** Verification of the recombined BAC and successful transgenesis by PCR. The heterozygously transmitted allele is indicated by +/T. **C)** Differential interference contrast (DIC) and fluorescence images of primary dermal lymphatic endothelial cells (pdLECs) from *Vegfr3*-tdTomato transgenic mice following magnetic bead-based isolation and enrichment. Images were captured in passage 1. Magnetic beads appear a round black spheres. Scale bars = 50 μm.

clone RP23-58E13 harbouring the Flt4 transgene by Red/ET recombineering. The expression cassette utilized the VEGFR-3 initiating ATG located in the first exon and replaced the remainder of the exon, practically putting tdTomato under the control of *Flt4* promotor elements (Fig 1A). Linearized BAC DNA of 214.464 kb size was used for pronuclear injection and yielded 74 mice, of which 14 had integrated the transgene, as verified by genomic PCR analysis (Fig 1B). To probe for tdTomato expression, the founders were mated to wildtype C56Bl/6 mice and we isolated and cultured endothelial cells from the dermis using anti-CD31-coated magnetic beads. The resulting primary cell cultures were comprised of a mixture of endothelial cells (ECs) of blood and lymphatic origin. The observed patches of tdTomato expression were expected, since only LECs should express consistently high levels of the

*Vegfr3*-driven tdTomato reporter (Fig 1C). In total we identified three individuals exhibiting a particularly bright tdTomato fluorescence i.e. expression.

For the subsequent establishment of a reporter mouse line, stably inheriting the transgene, we elected the founder displaying the highest relative brightness of tdTomato fluorescence. The female founder transmitted the transgene with Mendelian distribution and in all subsequent generations, *Vegfr3*-tdTomato mice were fertile and appeared healthy, without detectable abnormalities.

## The tdTomato fluorescence is associated with the lymphatic vessel network and co-localizes with VEGFR-3 protein expression

To test the expression of tdTomato in different lymphatic vessel beds, we analysed freshly isolated and unfixed organs from adult transgenic mice (Fig 2A–2F, 2I and 2J) and pups at postnatal day 5 (P5, Fig 2G, 2H and 2K–2T) using a fluorescence stereomicroscope and an inverted fluorescence microscope. Lymph vessels were easily identifiable due to the characteristic pattern of tdTomato expression in all analysed tissues including the diaphragm, lung, ear skin, intestinal mucosa, mesentery, mesenteric LN and heart (Fig 2).

We further investigated PFA fixed, wholemount immunostained tissues by CLSM to confirm co-expression of tdTomato with VEGFR-3 and the established lymphatic marker PROX1. Since fixation reduces the brightness of fluorescent proteins, we used an anti-RFP staining to visualize the tdTomato signal. In this analysis, all tdTomato-positive LVs were also found to express the VEGFR-3 protein (Figs 3A–3C and 4A–4C), verifying the lymphatic identity of the tdTomato-positive vessels and demonstrating faithful *Flt4* promoter-driven expression of the transgene. As expected, the tdTomato reporter was absent from CD31-positive, but VEGFR-3-negative blood vessels (Figs 3A, 3B and 4A–4C). While in adult mice VEGFR-3 is largely restricted to LVs, selected blood endothelial cells (BECs) have been shown to express VEGFR-3. These include tip cells of the postnatal retina [18] as well as liver sinusoidal endothelial cells [34] and we therefore investigated *Vegfr3*-tdTomato expression in these tissues (S1 Fig). In the angiogenic, sprouting front of the retina, tdTomato expression was absent although VEGFR-3 protein expression was confirmed by immunofluorescence staining (S1A Fig). In addition, immunostained vibratome sections of the adult liver showed prominent expression of tdTomato in LVs around the portal veins, while only a weak signal was detectable in VEGFR-3-positive sinusoids (S1B Fig).

Taken together these data show that in the newly generated *Vegfr3*-tdTomato mouse Tomato expression was readily detectable in LECs, in which it faithfully recapitulated VEGFR-3 expression.

## The fluorescent reporter tdTomato is expressed in both capillary and collecting lymphatic vessels of various tissues

Next, we wondered if in *Vegfr3*-tdTomato mice the fluorescent reporter was expressed in capLV and colLV, which both express VEGFR-3. In contrast to VEGFR-3, the hyaluronan receptor LYVE1 specifically marks the oak-leaf shaped LECs of capLVs. In the dermal LVs of the ear and the LVs of the diaphragm, blind-ending LYVE1-positive capLVs are particularly prominent. In immunostained wholemount preparations of both tissues, we readily noted co-localization of tdTomato and LYVE1 expression in capLVs during CLSM analysis (Fig 3A–3C). Despite the presence in all capLVs and colLVs vessels, we noted some heterogeneity in the tdTomato staining intensity between individual LECs within a vessel. To follow this observation further and to probe for a possible expression mosaicism, we analysed isolated cells from the lung of *Vegfr3*-tdTomato transgenic mice by flow cytometry (S2 Fig). Indeed, when

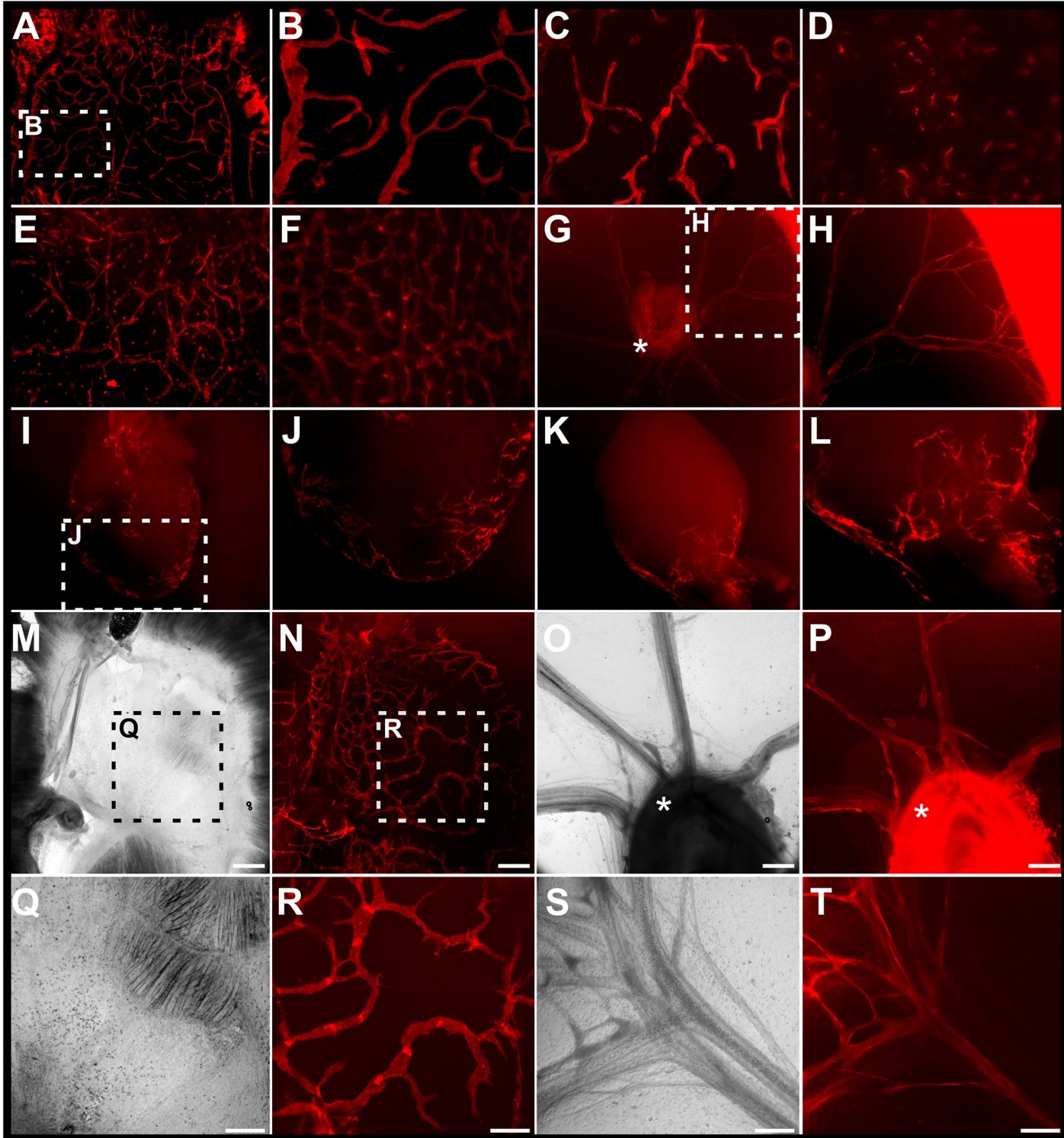

**Fig 2. Characterization of the reporter expression in *Vegfr3*-tdTomato transgenic mice.** Tissues prepared from adult mice (A-F, I-J) and pups at postnatal day 5 (G-H, K-T) were analysed for lymphatic expression of the *Vegfr3*-tdTomato transgene in a fluorescence stereo dissection microscope (A-L) and an inverted fluorescence microscope (M-T). Distinct lymphatic networks were detected in various adult tissues including the diaphragm (A-C), lung (D), ear dermis (E), intestinal mucosa (F), and heart (I-J) as well as postnatal tissues including the mesentery (G-H, O-P, S-T), mesenteric lymph nodes (G, O-P), heart (K-L) and diaphragm (M-N, Q-R). White asterisk denotes the position of the LN. The white and black boxed areas are magnified in the indicated panels. Scale bars = 400 μm (M-P) and 200 μm (Q-T).

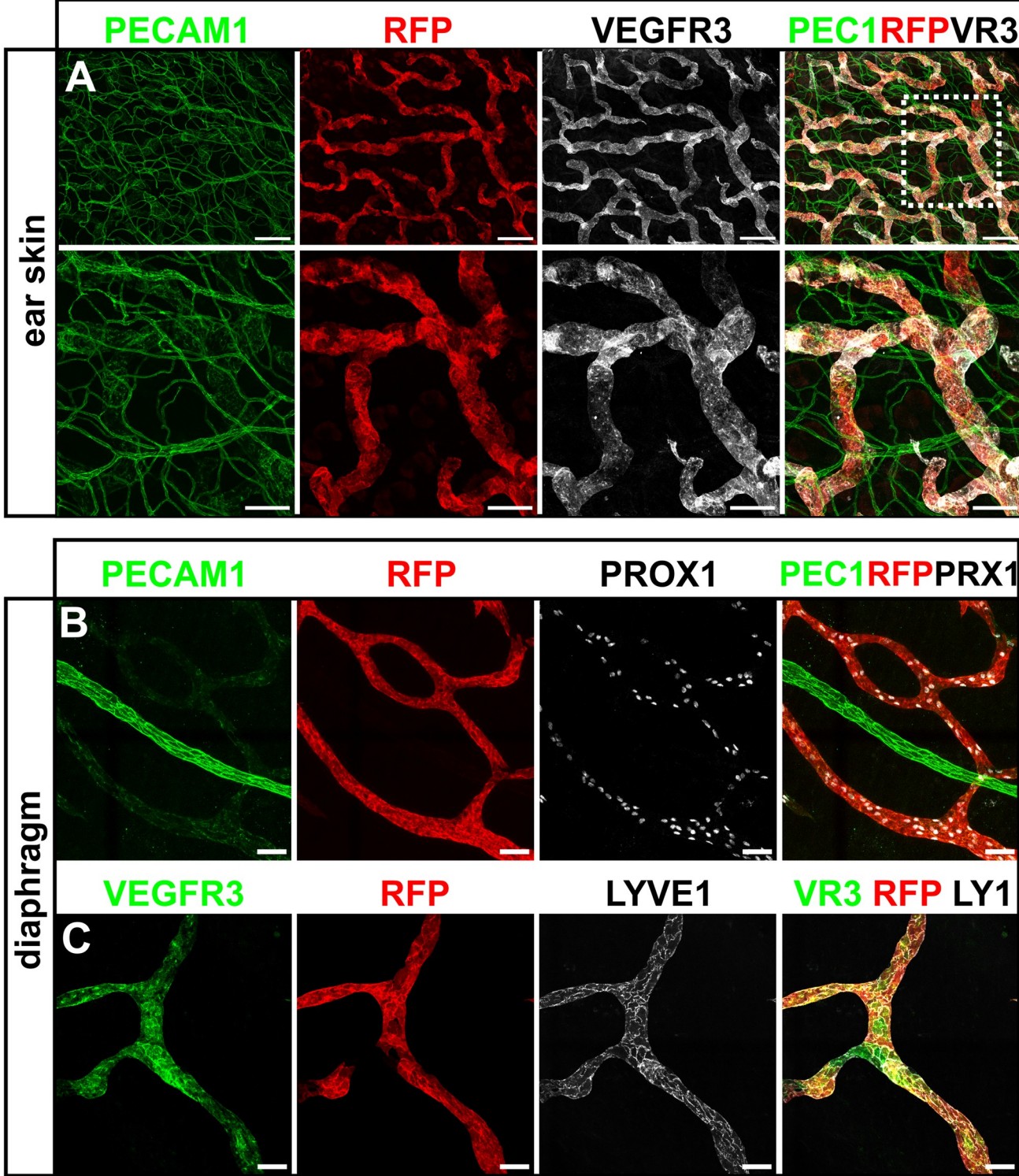

**Fig 3. Co-localization of the transgene-derived *Vegfr3*-tdTomato signal with bona fide lymphatic vessel markers in the dermis of the ear and diaphragm. A-C)** Maximum intensity projection (MIP) of representative confocal tile-scans from wholemount immunostained ear skin (A) and diaphragm (B-C) of adult *Vegfr3*-tdTomato transgenic mice. Stained antigens are indicated above each panel. The white boxed area in (A) is magnified in (B). Scale bars = 200μm (A), 100μm (A, magnification), 50μm (B,C).

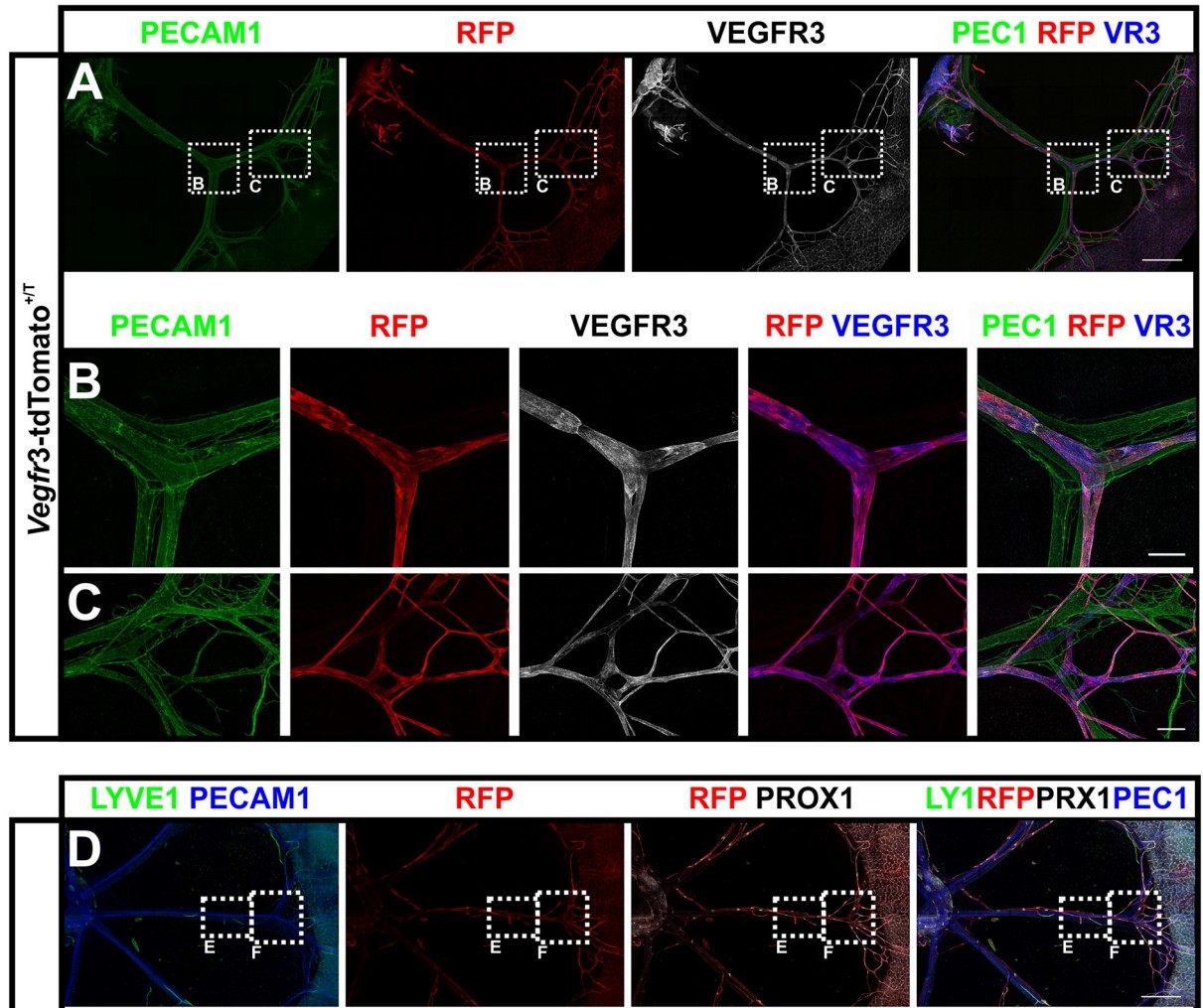

**Fig 4. Lymph vessel-specific expression of the *Vegfr3*-tdTomato transgene in the mesentery. A-F)** Wholemount immunostained preparations of mesenteries from *Vegfr3*-tdTomato transgenic pups at P5. Shown are MIPs of multi-tile z-stacks stained for the antigens depicted in colour above each panel. Magnifications of the areas boxed in white dashed lines in (A) and (D) are shown in the indicated panels (B,C and E,F). RFP staining identified transgene expression in lymphatic vessels. Yellow arrow heads indicate formation of semilunar valves. Scale bars = 1000 μm (A, B), 200 μm (B-C, E-F).

we gated on LECs, characterized by CD31 and Podoplanin (PDPN) expression, we found that only 2.83% of the double positive LECs of the lung showed tdTomato expression (S2E Fig). This value improved when we exclusively considered the PDPN$^{high}$ population, where 40% expressed tdTomato (S2F Fig).

Postnatal mesenteric LVs are prominently comprised of colLVs, which do not express LYVE1 but retain VEGFR-3, PROX1 and CD31 expression. Mesenteric colLVs strongly expressed tdTomato, which co-localized with the before-mentioned lymphatic markers (Fig 4). We identified newly forming intraluminal valves based on high PROX1 expression in the nascent semilunar leaflets (Fig 4E and 4F, yellow arrow heads). Collectively, these data confirm the lymphatic identity of tdTomato positive vessel structures and demonstrate consistent tdTomato expression in both capLVs and colLVs at different developmental stages.

## The *Vegfr3*-tdTomato mouse is a tool for two-photon microscopy

Presently, CLSM is probably the most widely used microscope modality in biomedical research, due to it low penetration depth and high energy transfer into tissue it is widely used for post mortem analysis of immunostained specimen. For intravital imaging or the analysis of deep tissue layers, 2P-LSM has become an established, yet not widely used technology, which may partly be due to relatively high costs, technological complexity and significantly lower optical resolution compared to CLSM. Significant advantages of 2P-LSM result from the near infrared excitation light used in this modality and include relatively deep tissue penetration ranging from few hundred micrometres to nearly one millimetre depending on the tissue investigated and the lower phototoxicity as compared to visible light [35].

Therefore, we were interested to demonstrate the suitability of our *Vegfr3*-tdTomato reporter mouse line for intravital or deep tissue imaging using 2P-LSM. As shown in Fig 5A, tdTomato is optimally excited at a wavelength of 1100 nm [25, 36]. *Ex vivo* 2P-LSM analysis of skin from *Vegfr3*-tdTomato foetuses (E14.5), in which the blood vessels had been contrasted with anti PECAM-1 antibodies revealed the tdTomato-expressing lymphatic vessel plexus and the stained blood vasculature up to a depth of 200 μm (Fig 5B). Also, in adult tissues, like the diaphragm visualization of tdTomato expressing LECs was readily possible. The label-free second-harmonic generation signals originating from the muscle myosin of the skeletal muscle provided anatomical positioning (Fig 5C). In the somewhat denser adult diaphragm, imaging up to a tissue depth of around 150 μm was possible. Taken together these results show the applicability of our *Vegfr3*-tdTomato reporter mouse for deep tissue imaging using 2P-LSM.

## The *Vegfr3*-tdTomato reporter model is also suitable for optical clearing and 3-dimensional visualization of lymphatic vessel networks by light-sheet fluorescence microscopy

Imaging of large tissue volumes (up to several cm$^3$) requires the use of specialized modalities in particular light sheet fluorescence microscopes (LSFM) are well suited for this task. LSFM requires optical clearing to reduce light scattering. Because most clearing procedures reduce the brightness of reporter proteins, the remaining fluorescence is typically not bright enough for imaging of large tissue volumes even cleared tissue samples. We tested *Vegfr3*-tdTomato reporter tissues using different optical clearing protocols to probe the suitability of this strain for large volume imaging. Organic solvent-based clearing protocols offer excellent tissue transparency but quench protein fluorescence virtually completely [37]. Therefore, we used whole-mount immunostaining with an RFP antibody to counterstain all tissue specimen derived from a *Vegfr3*-tdTomato pup at P5 that were subsequently cleared following the BABB protocol (Fig 6). 3D-volume rendering of image stacks was performed using the open source

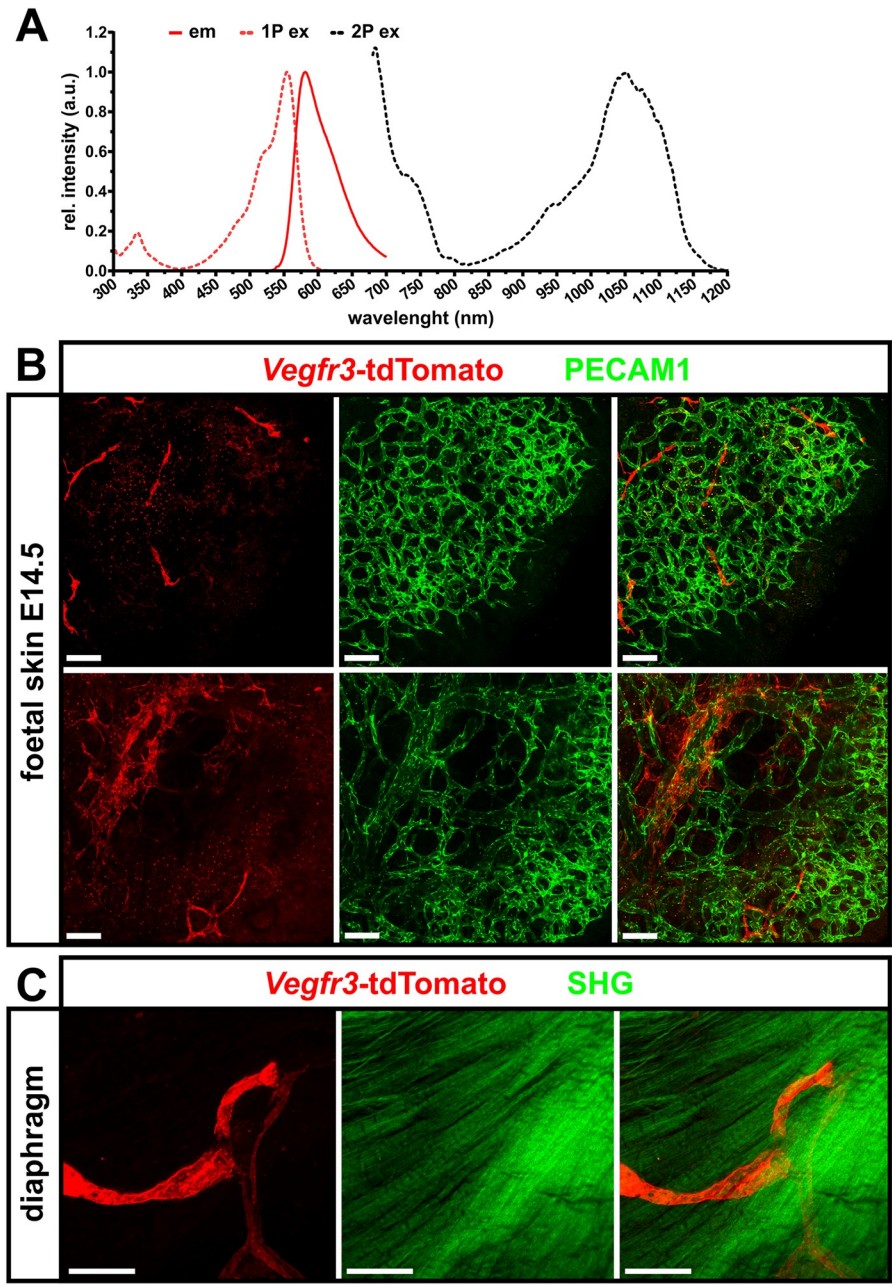

**Fig 5. 2-Photon laser scanning microscopy images of lymphatic vessels from transgenic *Vegfr3*-tdTomato reporter mice. A)** Spectral characteristics of the tdTomato fluorophore [33, 36]. 1P, single photon excitation spectrum, 2P, two photon excitation spectrum, em, emission spectrum. **B)** Expression of *Vegfr3*-tdTomato reporter construct in the superficial lymphatic vasculature of foetal skin (E14.5) imaged by 2P-LSM. The blood vasculature was contrasted with Alexa Fluor™ 647-labelled anti PECAM1 antibodies. **C)** Confirmation of *Vegfr-3* promoter driven expression of the fluorescent reporter protein tdTomato in lymphatic vessels of the adult thoracic diaphragm. Due to its non-centrosymmetric biomolecular organization skeletal muscle of the diaphragm generates intense second harmonic generation (SHG) signals, that are visualized as green, repetitively striated signal. Shown are MIPs of 200 μm (B) and 150 μm (C) z-stacks. Scale bars = 50 μm.

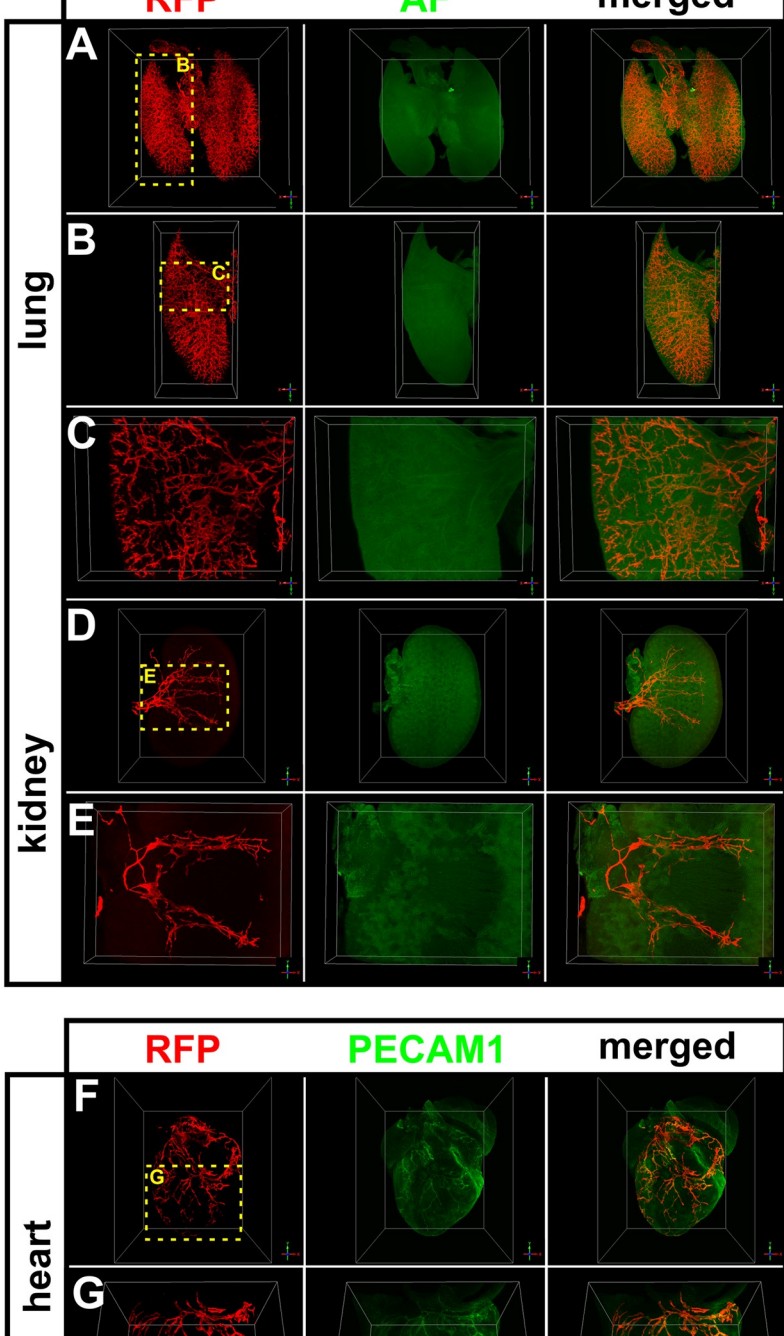

**Fig 6. Light sheet microscopic visualization of the lymphatic vasculature in transgenic *Vegfr3*-tdTomato reporter mice after organic solvent-based tissue clearing. A-E)** Volume reconstruction of lymphatic vessels in the lung (A-C) and kidney (D+E) of *Vegfr3*-tdTomato reporter mice at P5. Immunostaining for RFP identified expression of the *Vegfr3*-tdTomato transgene. Endogenous tissue autofluorescence (AF) allowed contrasting of the overall organ volume. **F+G)** 3D visualization of cardiac lymphatic vessels by immunostaining for RFP indicating expression of the *Vegfr3*-tdTomato transgene. Blood vasculature was visualized by Alexa Fluor™ 647-labelled anti PECAM1 antibodies. The yellow boxed areas are magnified in the indicated panels.

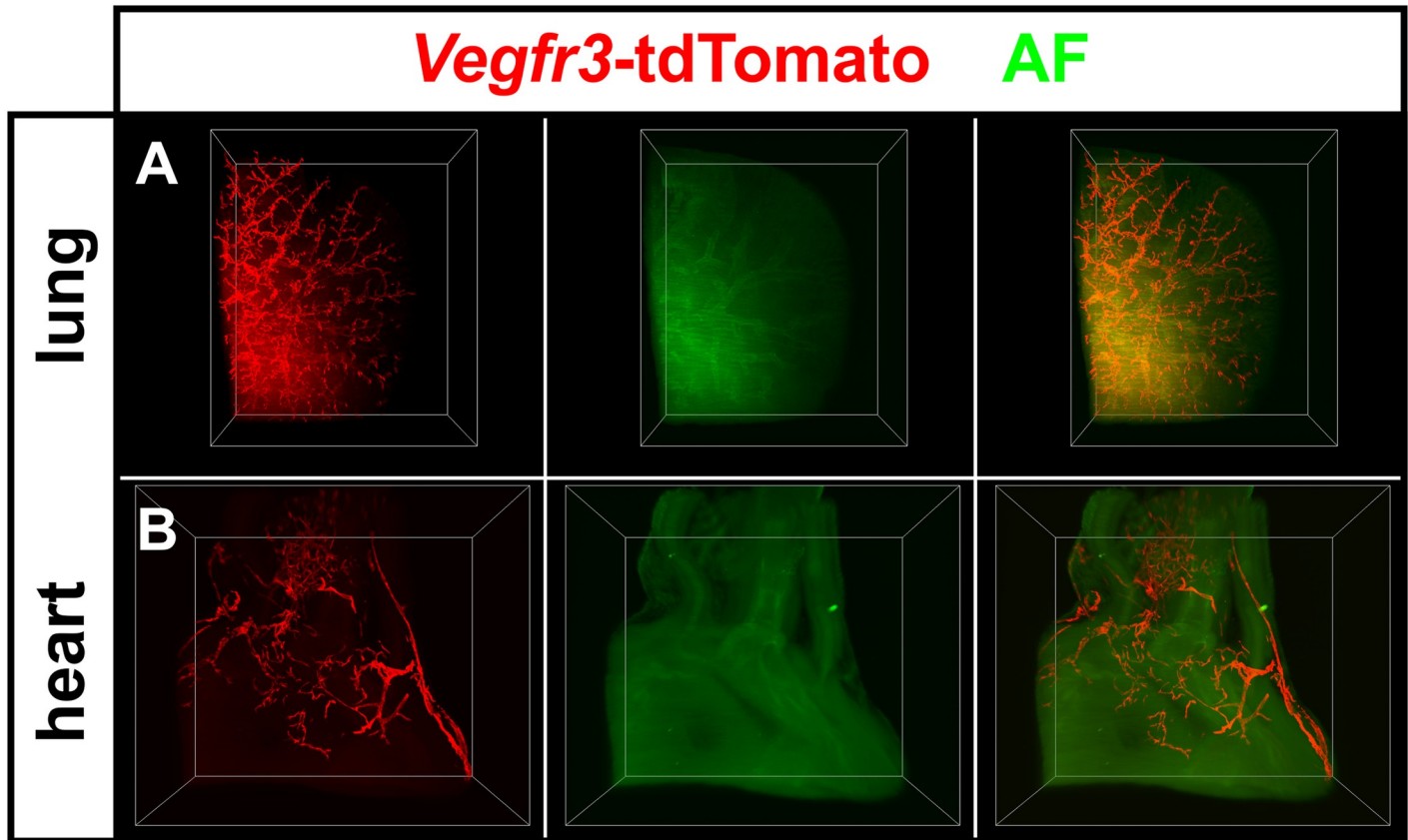

**Fig 7. Light sheet microscopy of CUBIC-cleared tissues visualized *Vegfr3*-tdTomato transgene expression in lymphatic vessels of the mouse lung and heart. A +B)** 3D visualization of tdTomato positive lymphatic vessels in the lung (A) and heart (B) of *Vegfr3*-tdTomato reporter mice at P5. Tissue autofluorescence (AF) was used to contrast organ topography.

visualization software package Voreen [38, 39]. In the lung a highly branched lymphatic network was observed lancing both lobes to the level of the terminal bronchioles (Fig 6A–6C). In addition, prominent lymphatic vessels surrounded the trachea and primary bronchi (S1 Video). The organ volume was delineated via tissue autofluorescence in the green channel (Fig 6A–6C). A kidney explant allowed visualization of the developing renal lymphatic vasculature. Lymph vessels, that start to pervade the renal cortex, follow interlobar blood vessels towards the renal pelvis, where they run into large hilar collecting lymph vessels located adjacent the major renal arteries and veins (Fig 6D and 6E, S2 Video). Finally, we visualized the developing LVs in the mouse heart, which as previously described [40]. LVs are located particularly in the epicardium but hardly invade into the myocardium as described for larger mammals. In addition, LVs form a dense network around the root of the aorta (Fig 6F+6G, S3 Video).

To overcome the limitations associated with wholemount immunostaining and to minimize the fluorescence loss of tdTomato due to tissue clearing as much as possible, we applied the hydrophilic tissue clearing protocol CUBIC (Fig 7) [41]. Here fluorescence of tdTomato was retained allowing the direct visualization of the lymphatic vasculature of the lung's right lobe and the heart without need for counterstaining. Also in CUBIC-cleared tissue green autofluorescence was used to delineate organ outline and volume (Fig 7A and 7B). Collectively, these results show that large volume imaging of LVs in *Vegfr3*-tdTomato reporter mice was

possible using organic solvent-based or hydrophilic tissue clearing methods. However, for organic solvent-based clearing wholemount immunostaining was mandatory to compensate the loss of protein fluorescence.

## Discussion

We report the generation and characterization of a *Vegfr3*-tdTomato reporter mouse line for imaging the lymphatic vasculature by multiple modalities. Although not formally proven in this study, the *Vegfr3*-tdTomato line should also be suitable for combination of microscopic imaging with molecular or whole-body imaging in multi-modal approaches. The tdTomato reporter showed bright, homogenous and consistent red fluorescence in LECs of all LVs investigated in this study and was successfully applied in a wide range of microscope modalities, including fluorescence stereomicroscopy, CLSM, 2P-LSM and LSFM. Further, the reporter can be exploited for the isolation of LECs from tissues and can be combined with other fluorescent proteins for multi-colour analysis.

For generation of the transgenic line, we inserted a cDNA expression cassette encoding a membrane-tagged version of tdTomato into the BAC RP23-58E13 that harbours 214kB of mouse genomic sequence from chromosome 11 including the *Flt4* coding and presumably most regulatory sequences. The expression cassette was inserted by Red/ET recombineering such that tdTomato utilizes the initiating ATG of the VEGFR-3 and the cDNA replaces the remainder of Exon1. We chose tdTomato because it is among the brightest red fluorescent proteins [33]. Consequently, we expected our reporter to allow deeper imaging compared to other available LEC-specific reporter lines [22, 23, 25–28, 31, 42]. Furthermore, in contrast to previously published GFP-based lymphatic reporters [29, 30, 32] the red-shifted excitation and emission spectrum of tdTomato should aid in avoiding abundant green tissue autofluorescence (Fig 5A). In addition, combination with other especially green fluorescent reporter mouse lines is possible to simultaneously visualize for example distinct vascular beds by crossbreeding with lines specifically labelling blood ECs [43–45].

Following pronuclear microinjection, analysis of the transgenic offspring indicated that tdTomato had been successfully subjected to transcriptional control of the *Flt4* gene, hence restricting the expression to LECs but not BECs after the onset of embryonic LV formation. To establish the *Vegfr3*-tdTomato reporter mouse line, we bred the founder displaying the brightest and most uniform tdTomato-fluorescence in all analysed LV beds. The transgene genomic integration site tends to be of minor importance in BAC transgenesis, indeed locus-specific influences bear the danger to interfere with faithful control through the transgene promotor, which in our *Vegfr3*-tdTomato line appeared however to be maintained. A possible explanation for the superior tdTomato expression might be the integration of multiple copies, very likely in the same integration site, as we did not note segregation of transgenes in subsequent generations.

In all anatomical locations analysed, tdTomato expression was restricted to the lymphatic vasculature, while absent from blood vessels. Several proteins including PROX1, PDPN, VEGFR-3 and LYVE1 have in recent years been used as markers to identify or isolate LECs, however, none of these is exclusively expressed in LECs. LYVE1 expression in adult mice is besides inflammatory cells largely restricted to capLVs and hence not suitable for reporter expression [46]. Besides LECs, PROX1 is expressed in a wide range of non-endothelial cell types (e.g. hepatocytes [47], cardiomyocytes [48] or neural cells of the retina [49]), while PDPN displays an even wider expression pattern, including various types of fibroblasts, epithelial and hematopoietic cells [50]. An advantage of *Vegfr*-driven reporter gene expression, is the more EC-restricted VEGFR-3 expression, which in adult mice in addition to LECs is

limited to a small subset of BECs in the angiogenic front of the developing retina [18], sinusoidal endothelial cells of the liver [34] and BECs in tumors [51]. Interestingly, in our *Vegfr3*-tdTomato mouse, VEGFR-3-expressing was not detectable on retinal tip cells and only weakly on liver sinusoidal endothelial cells. A hierarchical network of LVs was detectable and co-localization with lymphatic-specific markers in capLVs and colLVs was retained in wholemount preparations. The *Vegfr3*-tdTomato transgene recapitulated largely the endogenous VEGFR-3 expression in LECs and the plasma membrane localization aided visualization of tubular LV structures and distinctly delineated the shape of single LECs. In our wholemount stained samples, we noted some heterogeneity of the tdTomato staining intensity, suggesting that marker expression was patchy. Flow cytometric verification indeed detected a substantial proportion of tdTomato-negative cells in the CD31, PDPN double positive population that was least in the CD31, PDPN^High cells. This, together with the absence or only sporadic marker expression in retinal tip cells and liver sinusoidal endothelial cells suggests that the Vegfr3-tdTomato mouse does not report weak VEGR-3 promotor activity, which nevertheless results in an unexpected even beneficial specificity for LVs. Reasons may reside in the lack of distant regulatory elements in the promotor-containing BAC used to generate this mouse model or in the inability of the chosen fluorescent protein to report minute expression levels. Given its high brightness and fast maturation time tdTomato was the protein of choice. Certainly following immunostaining, we were able to visualize all LVs that would be identifiable on the basis of the before mentioned marker proteins.

Since dehydration and lipid extraction during optical tissue clearing with organic solvents efficiently quenches protein-based fluorescence, we used wholemount immunostaining with an anti-RFP antibody to detect tdTomato expression, which allowed us to visualize deep LVs in large tissue volumes [52]. While providing bright signals with a high signal to noise ratio, wholemount immunostaining remains limited by potential inaccessibility of the tissue to antibody staining. To overcome this obstacle, we applied the CUBIC tissue clearing protocol, which is based on detergent-mediated lipid extraction in hyper-hydrated samples [41]. This protocol retained the tdTomato fluorescence in transgenic tissues to a degree where again three-dimensional visualization of deep LVs was possible. For the analysis of complex vascular structures in large tissue volumes LSFM followed by digital image reconstruction is the most suitable imaging modality, which allowed us to reconstruct the complexity of the lymphatic vasculature in whole organs with a level of details that has not been shown to this extend and quality earlier. Finally, tissue clearing is not possible when intravital imaging of tdTomato in live animals is attempted. We demonstrated the suitability of our reporter mouse by 2P-LSM on unfixed, freshly isolated samples, where the fluorescence of LV-associated tdTomato was readily detectable up to a depth of 300 μm at an excitation wavelength of 1100 nm.

## Conclusion

The *Vegfr3*-tdTomato reporter mouse strain that we have described in this study showed reliable labelling of lymph vessels in all analysed tissues including the dermis, mesentery, lymph nodes, intestinal mucosa, diaphragm, heart, lung, liver and kidney.

The model is applicable to tissue clearing, the CUBIC protocol will preserve the endogenous fluorescence of tdTomato. The line was suitable for a wide range of different imaging techniques and it allowed us to visualize the, so far not reported, complex architecture of the deep lymphatic plexus in various organs. Moreover, our reporter opens a number of important applications in intravital microscopy as it is photostable and bright. It is therefore a useful and advantageous tool for future dynamic explorations of the lymphatic vasculature.

## Material and methods

### Generation of a *Vegfr3*-tdTomato reporter mouse

We have flanked a cDNA, coding for the membrane-tagged RFP-derivate tdTomato-CAAX followed by an ampicillin-resistance cassette for positive selection with homology regions of the murine *Flt4* gene. Targeting vectors were sequenced to verify correct insertion of the DNA fragments. We used Red/ET recombineering in *E. coli* DH10B to generate a modified bacterial artificial chromosome (the BAC clone RP23-58E13, harbouring the Flt4 gene within 214kB of genomic sequence of mouse chromosome 11). The construct was designed such that the VEGFR-3 start codon became the tdTomato start codon. Linearized DNA was used for pronuclear injection into fertilized zygotes, which were implanted into pseudo-pregnant C57Bl/6 females. Transgenic founders were identified by genomic PCR analysis (PCR1: fwd 5'– GACAACAACATGGCCGTCA–3', rev 5'–CTTGTACAGCTCGTCCATGC–3' and PCR2: fwd 5'– GCTCTCACTCCCAGCCTAG–3', rev 5'–ACTCTTTGATGACGGCCATGTT–3') and were subsequently screened for *Vegfr3* promotor-driven tdTomato expression by stereomicroscopy of the dermal lymphatic vascular plexus in ear skin biopsies. One of three founders was selected for line establishment, based on the brightness of tdTomato fluorescence.

### Animal experiments

Mice were kept under conventional conditions in IVC-cages and ventilated racks at 22˚C and 55% humidity with a light-dark cycle of 14:10 h. Transgenic *Vegfr3*-tdTomato mice were analysed at the age of postnatal day 5 (P5) to 21 weeks. Foetuses were analysed at developmental stage of E14.5. Embryonic staging was determined by the day of the vaginal plug (E 0.5). Wild-type littermate mice served as controls. In all experiment's mice were sacrificed by cervical dislocation. All Procedures involving animals were carried out in strict accordance with the German animal protection legislation (Tierschutzgesetz und Tierschutzversuchstierverordnung). The protocol was approved (84–02.04.2016.A218) by the Committee on the Ethics of Animal Experiments of the Landersamt für Natur, Umwelt und Verbraucherschutz (LANUV).

### Antibodies

The following antibodies were used: rabbit polyclonal anti-human PROX1 (ReliaTech, 102-PA30), goat polyclonal anti-human PROX1 (R&D Systems, AF2727), rat monoclonal anti-mouse PECAM-1 (clone 5D2.6 and clone 1G5.1 [53]), goat polyclonal anti-mouse LYVE1 (R&D Systems, AF2125), rat monoclonal anti-mouse LYVE1 (clone 223322, R&D Systems; MAB2125), goat polyclonal anti-mouse VEGFR-3 (R&D Systems, AF743), rabbit polyclonal anti-RFP (Rockland, 600-401-379), rat monoclonal anti-CD31 FITC (clone 390; Invitrogen, Cat# RM5201), Syrian Hamster anti-Podoplanin eFluor®660 (clone 8.1.1; eBioscience, Cat# 50-5381-82) and Isolectin GS-IB4 Alexa Fluor™ 647 conjugate (Invitrogen, Cat# I32450). Secondary Alexa-dye labelled antibodies were obtained from Life Technologies.

### Isolation of murine lung cells for flow cytometric analysis

Cells were isolated from the lung of *Vegfr3*-tdTomato mice. Tissues were enzymatically digested in 1 mg/ml Collagenase A solution for 30 min at 37˚C. Single cells were collected and stained for 30 min with dye-conjugated primary antibodies diluted in FACS buffer (3% fetal calf serum in PBS). After antibody staining, the cells were analyzed on a FACSAria IIIu cell sorter (BD Biosciences) using an 85 μm nozzle. Cells were gated based on FSC and SSC to exclude cellular debris and doublets and on DAPI exclusion for viability. Cells were gated on

the basis of CD31 and PDPN double positivity and then analysed for tdTomato expression. Data analysis was done using FlowJo software (LLC).

## Isolation and culture of dermal endothelial cells

Primary dermal endothelial cells (ECs) were isolated from the tail skin of transgenic mice as described previously [54]. In brief, tail skin fragments were digested with 5% dispase/PBS to remove the epidermis from the dermis. Dermal fragments were then transferred into 1 mg/mL Collagenase A solution to maintain a single cell suspension. Single cells were incubated with magnetic dynabeads (coated with sheep anti-rat IgG) coupled to anti-CD31 antibody (clone mec13-3). CD31-positive ECs of both lymph- and blood vessel origin were enriched by magnetic separation. For cultivation of lymphatic endothelial cells (LECs) preselected batches of ECGS (BT-203, Alfa Aesar) that favour the expansion and maintenance of lymphatic endothelium were used. LECs were maintained in Dulbecco's Modified Eagle's Medium (DMEM) supplemented with 20% FCS, 2 mM L-glutamine, 100 μg/mL penicillin/streptomycin, 1% sodium pyruvate, 0,1 mM non-essential amino acids, 20 μg/mL ECGS, 50 μg/mL Heparin, 50 μM β-mercaptoethanol and cultured on 0.4% gelatine coated petri dishes or multi-well plates at 37˚C, 10% $CO_2$ and 95–100% humidity. Enriched LEC cultures in passage 1 were used for fluorescence imaging.

## Wholemount staining of embryonic and adult tissues

Tissues were fixed in 4% PFA for 2 h and washed with PBS. Subsequently, samples were permeabilized in 0.5% Triton X-100 in PBS, blocked in PermBlock solution (3% BSA, 0.1% Tween-20 in PBS) and stained with the listed primary antibodies at 4˚C. Following three washing steps with PBS-T (0.1% Tween-20), tissues were incubated in secondary antibodies labelled with Alexa-dyes and mounted with Mowiol.

## Optical tissue clearing

Optical clearing was described previously [41]. Briefly, prior fluorescence light sheet imaging, postnatal wholemount stainings were embedded in 2% low-melting point agarose. For organic solvent-based clearing the samples were dehydrated in increasing concentrations of methanol (50%, 70%, 99.5%, 99.5% methanol (v/v), each step at least 1 hour) and then optically cleared in a benzyl alcohol:benzyl benzoate solution (BABB; ratio 1:2) for 4h.

For CUBIC-based tissue clearing samples were incubated in reagent-1A up to 4 days. while reagent-1A was replaced every 24 hours. Afterwards samples were washed several times in PBS and were then incubated reagent-2 until final transparency. Before Imaging the samples were immersed in silicon oil (Sigma, 175633) for at least 1 hour.

## Microscopy

**Widefield fluorescence microscopy.** Imaging of murine tissue samples and LEC cultures was conducted on a Nikon Eclipse Ti2 inverted fluorescence microscope (Nikon, Japan) using a 10X Plan Fluor (NA 0.3, WD 15.20 mm) objective and appropriate emission filter sets for Cy3 (577/25 nm). The system is equipped with a SPECTRA X light engine® (Lumencor, USA) for excitation.

**Stereomicroscopy.** Unfixed and unstained *Vegfr3*-tdTomato mice were analysed with a Leica Stereomicroscope MZ16F coupled to a digital camera (Hamamatsu C4742-95). Volocity software (Perkin Elmer) was used for image acquisition and processing.

**Confocal laser scanning microscopy and image processing.** Confocal images were captured using an LSM 880 confocal microscope (Carl Zeiss; 10x, NA = 0.45; 20x, NA = 0.8; 40x water, NA = 1.2; 63x oil, NA = 1.4). Microscopy data were recorded and processed with ZEN Pro software (Carl Zeiss). All confocal images represent maximum intensity projections of z-stacks of either single tile or multiple tile scan images. Mosaic tile-scans with 10% overlap between neighbouring z-stacks were stitched in ZEN software. Confocal single and multi-tile-scans were processed in Fiji [55]. If necessary, adjustments to brightness, contrast and intensity were equally accomplished for individual channels and compared data sets.

**Light sheet fluorescence microscopy and 3D-image processing.** Optically cleared samples were imaged on a LaVision BioTec Ultramicroscope II (LaVision BioTec, Bielefeld, Germany) equipped with an Olympus MVX10 Zoom Microscope Body (Olympus, Tokyo, Japan) allowing an optical magnification range from 1.26x to 12.6x and an NA of 0.5. An NKT SuperK (Power SK PP485) supercontinuum white light laser served as excitation light source. For excitation and emission detection of specific fluorophores custom band-pass filters (excitation 470/40, 577/25 or 640/30 nm; emission 525/50, 632/60 or 690/50 nm) in combination with an Andor Neo sCMOS Camera. For image acquisition, Z-steps 3 μm were chosen. 3D reconstruction and analysis of ultramicroscopy stacks were performed by using the volume rendering software Voreen [39, 56].

**Two-photon laser scanning microscopy.** For *ex vivo* imaging of embryonic dermal lymphatic vessels and adult diaphragm tissue a LaVision TriM Scope II microscope (LaVision Bio-Tec, Bielefeld, Germany) was used with a water dipping objective (CFI-Apo LWD 25XW, NA = 1.1, WD = 2.0 mm, Nikon). This system is equipped with a Chameleon Discovery ultra-fast tuneable, which is used to pump an optical parametric oscillator (OPO that generates wavelengths greater than 1000 nm and a Chameleon XR Femtosecond Ti:Sapphire laser (Coherent, Santa Clara, USA) to generate wavelengths up to 850 nm. Either second harmonic generation signal (SHG) or Alexa Fluor™ 647 were simultaneously excited at 1100 nm with tdTomato.

## Supporting information

**S1 Fig. Expression of the *Vegfr3*-tdTomato transgene in the developing retina and the liver.** MIPs of representative confocal tile-scans from immunostained wholemount preparations of the developing retina (**A**) and vibratome sections of the liver (**B**) of postnatal *Vegfr3*-tdTomato transgenic mice. Stained antigens are indicated above each panel. Yellow arrow heads in B indicate VEGFR-3 and RFP double positive sinusoids. PV = portal vein. Scale bars = 300 μm (A) and 100 μm (B).
(TIF)

**S2 Fig. FACS analysis of isolated cells from the lung of *Vegfr3*-tdTomato mice.** Representative FACS plots and gating scheme are shown. Isolated cells immunostained for the pan-endothelial cell marker CD31 and the common lymphatic surface marker PDPN were gated on FSC and SSC (**A**) to exclude cellular debris and doublets (**B**), and on staining with DAPI (**C**) to exclude dead cells. Different cell populations co-expressing CD31 and PDPN (**D**), were analysed for tdTomato positivity (**E** and **F**).
(TIF)

**S1 Video. Volume imaging and 3D reconstruction of the lymphatic vessel system in the lung of a 5-day old mouse pup.** The explanted organ was fixed and wholemount immunostained with anti RFP antibodies. For tissue clearing the sample was dehydrated and delipidated by stepwise incubation in increasing concentrations of methanol followed by refractory index

matching in BABB. Image stacks acquired by LSFM were digitally rendered using the open source visualization software package Voreen. Shown is a consecutive slice view throughout the sample, followed by volume reconstruction of the vessel bed, a 360 degree turn and a zoom in to reveal details of the LVs (red). The organ volume is outline by tissue autofluorescence (green).
(MP4)

**S2 Video. Volume imaging and 3D reconstruction of the lymphatic vessel system in the kidney of a 5-day old mouse pup.** The explanted organ was fixed and wholemount immunostained with anti RFP antibodies. For tissue clearing the sample was dehydrated and delipidated by stepwise incubation in increasing concentrations of methanol followed by refractory index matching in BABB. Image stacks acquired by LSFM were digitally rendered using the open source visualization software package Voreen. Shown is a consecutive slice view throughout the sample, followed by volume reconstruction of the vessel bed, a 360 degree turn and a zoom in to reveal details of the LVs (red). The organ volume is outline by tissue autofluorescence (green).
(MP4)

**S3 Video. Volume imaging and 3D reconstruction of the lymphatic vessel system in the heart of a 5-day old mouse pup.** The explanted organ was fixed and wholemount immunostained with anti RFP antibodies. For tissue clearing the sample was dehydrated and delipidated by stepwise incubation in increasing concentrations of methanol followed by refractory index matching in BABB. Image stacks acquired by LSFM were digitally rendered using the open source visualization software package Voreen. Shown is a consecutive slice view throughout the sample, followed by volume reconstruction of the vessel bed, a 360 degree turn and a zoom in to reveal details of the LVs (red). The organ volume is outline by tissue autofluorescence (green).
(MP4)

**S1 Raw images.**
(PDF)

## Acknowledgments

We are grateful to Dietmar Vestweber for his support and provision of valuable reagents. We thank Barbara Waschk, Ludmila Kremer and Nannette Kümpel-Rink for expert technical support.

## Author Contributions

**Conceptualization:** Esther Redder, Nils Kirschnick, René Hägerling, Nils Rouven Hansmeier, Friedemann Kiefer.

**Data curation:** Esther Redder, Nils Kirschnick.

**Formal analysis:** Esther Redder, Nils Kirschnick, René Hägerling, Nils Rouven Hansmeier.

**Funding acquisition:** Friedemann Kiefer.

**Investigation:** Stefanie Bobe, René Hägerling, Nils Rouven Hansmeier, Friedemann Kiefer.

**Methodology:** Esther Redder, Nils Kirschnick, Stefanie Bobe, René Hägerling, Nils Rouven Hansmeier.

**Project administration:** Friedemann Kiefer.

**Resources:** Friedemann Kiefer.

**Supervision:** Friedemann Kiefer.

**Validation:** Esther Redder, Nils Kirschnick.

**Visualization:** Esther Redder, Nils Kirschnick, Stefanie Bobe.

**Writing – original draft:** Esther Redder, Nils Kirschnick, Friedemann Kiefer.

**Writing – review & editing:** Esther Redder, Nils Kirschnick, René Hägerling, Nils Rouven Hansmeier, Friedemann Kiefer.

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
