## [Decision Letter · Decision Letter 0]

8 Apr 2021

PONE-D-21-08111

Vegfr3-tdTomato, a reporter mouse for microscopic visualization of lymphatic vessel by multiple modalities

PLOS ONE

Dear Dr. Kiefer,

Thank you for submitting your manuscript to PLOS ONE. After careful consideration, we feel that it has merit but does not fully meet PLOS ONE’s publication criteria as it currently stands. Therefore, we invite you to submit a revised version of the manuscript that addresses the points raised during the review process.

The reviewers particular raise the minor points that in some images the tdTomato reporter does not perfectly match the staining with LYVE1 and/or VEGFR3. Both reviewers suggest to perform analysis of LECs by FACS for the reporter and the lymphatic markers mentioned above. Moreover, reporter expression in other cells and tissues, as well as in angiogenic sprouts in the retina that have been reported to express VEGFR3 should be mentioned and discussed. In general, the advantages and disadvantages of the presented reporter mouse models should be highlighted and discussed in more depth.

We look forward to receiving your revised manuscript.

Kind regards,

Stefan Liebner, PhD

Academic Editor

PLOS ONE

Journal Requirements:

5. Please amend your Methods section and Ethics Statement to provide details about methods of animal anesthesia and euthanasia.

6. Thank you for including your ethics statement:  "All animal procedures were performed according to relevant laws and institutional guidelines, were approved by the state animal ethics committee and were conducted with permission (84-02.04.2016.A218) granted by the Landesamt für Natur, Umwelt und Verbraucherschutz (LANUV) of North Rhine-Westphalia.".   

Please amend your current ethics statement to include the full name of the ethics committee that approved your specific study.

For additional information about PLOS ONE submissions requirements for ethics oversight of animal work, please refer to http://journals.plos.org/plosone/s/submission-guidelines#loc-animal-research  

7. We note that you have indicated that data from this study are available upon request. PLOS only allows data to be available upon request if there are legal or ethical restrictions on sharing data publicly. For more information on unacceptable data access restrictions, please see http://journals.plos.org/plosone/s/data-availability#loc-unacceptable-data-access-restrictions.

8. We note that you have included the phrase “data not shown” in your manuscript. Unfortunately, this does not meet our data sharing requirements. PLOS does not permit references to inaccessible data. We require that authors provide all relevant data within the paper, Supporting Information files, or in an acceptable, public repository. Please add a citation to support this phrase or upload the data that corresponds with these findings to a stable repository (such as Figshare or Dryad) and provide and URLs, DOIs, or accession numbers that may be used to access these data. Or, if the data are not a core part of the research being presented in your study, we ask that you remove the phrase that refers to these data.

Reviewers' comments:

Reviewer's Responses to Questions

**Comments to the Author**

1. Is the manuscript technically sound, and do the data support the conclusions?

Reviewer #1: Yes

Reviewer #2: Yes

2. Has the statistical analysis been performed appropriately and rigorously? 

Reviewer #1: N/A

Reviewer #2: Yes

3. Have the authors made all data underlying the findings in their manuscript fully available?

Reviewer #1: Yes

Reviewer #2: Yes

4. Is the manuscript presented in an intelligible fashion and written in standard English?

Reviewer #1: Yes

Reviewer #2: Yes

5. Review Comments to the Author

Reviewer #1: The authors describe here a novel BAC transgenic Vegfr3-tdTomato mouse line for studying the lymphatic vasculature. By using multiple imaging modalities (confocal, 2-photon, light sheet) they highlight the technical potential of the newly described line and its application for analysis of lymphatic vessel architecture in large tissue samples. The data are convincing, with excellent quality of imaging showing lymphatic vessel labelling in multiple tissues. The line will undoubtedly become a valuable resource in the field as it provides multiple benefits compared to existing lines: for example, a single transgene driven expression of tdTomato characterized by fast maturation and exceptional brightness as well as suitability for intravital / deep tissue imaging using 2-photon microscopy due to near infrared excitation. I have just a few suggestions for consideration:

Major comments:

1. It would be informative to show if the reporter recapitulates the endogenous expression of Vegfr3 in a subset of BECs, for example sinusoidal ECs (e.g. liver) or tip cells of angiogenic sprouts (e.g. developing retina).

2. In Fig 3, it seems that some VEGFR3+LYVE1+ LECs are not positive for tdTom (or anti-RFP staining in this case). This is particularly evident in Fig 3C but can also be seen in Fig 3A. Overall, RFP signal intensity seems to be heterogenous and not necessarily correlating with VEGFR3 staining intensity. Could the authors comment on this? Flow cytometric analysis of tdTom+ LECs could be useful.

Minor comments:

1. line 244: “… the epicardium but hardly invade into the myocardium are described for larger mammals.” I believe it is a typo and should be “… the epicardium but hardly invade into the myocardium as described for larger mammals.

Reviewer #2: Redder et al report generation of Vegfr3-tdTomato transgenic mice and demonstrate utility of such model for visualization of lymphatic vessels in different organs using a variety of imaging approaches, such as confocal, light sheet and two-photon microscopy. Overall, the results are clearly presented and the manuscript is well written. The model will be a useful addition to the arsenal of tools for characterization of lymphatic vessels, especially in adult animals and pathological conditions. I have only few comments as detailed below.

1. It seems that a proportion of LECs does not express the reporter (e.g. Figure 3 C), could the authors quantify the proportion of such tdTomato-negative LECs using FACS?

2. In addition to LECs, Vegfr3 expression has been reported in the tips of sprouting blood vessels in postnatal retina and tumors, in some fenestrated blood endothelial cells, immune cells and neural stem cells in subventricular zone of brain. Do the authors observe such pattern of expression in their model?

3. The authors should discuss in more detail advantages and disadvantages of their model in comparison to other existing lymphatic reporter strains. For example, expression of Prox1 in multiple non-endothelial cell types sometimes precludes use of Prox1- based reporters.

6. PLOS authors have the option to publish the peer review history of their article (what does this mean?). If published, this will include your full peer review and any attached files.

Reviewer #1: No

Reviewer #2: No

---

## [Author Response · Author response to Decision Letter 0]

8 Jul 2021

Detailed point by point response to the Reviewers' comments on the manuscript PONE-D-21-08111 by Redder et al. “Vegfr3-tdTomato, a reporter mouse for microscopic visualization of lymphatic vessel by multiple modalities”:

Review Comments to the Author

Reviewer #1: 

The authors describe here a novel BAC transgenic Vegfr3-tdTomato mouse line for studying the lymphatic vasculature. By using multiple imaging modalities (confocal, 2-photon, light sheet) they highlight the technical potential of the newly described line and its application for analysis of lymphatic vessel architecture in large tissue samples. The data are convincing, with excellent quality of imaging showing lymphatic vessel labelling in multiple tissues. The line will undoubtedly become a valuable resource in the field as it provides multiple benefits compared to existing lines: for example, a single transgene driven expression of tdTomato characterized by fast maturation and exceptional brightness as well as suitability for intravital / deep tissue imaging using 2-photon microscopy due to near infrared excitation. I have just a few suggestions for consideration:

We thank the Reviewer #1 for the positive feedback. Indeed, we believe our novel reporter line offers several advantages over already published reporter lines for the identification of lymphatic vessels and therefore will contribute to the future investigation of the lymphatic system.

Major comments:

1. It would be informative to show if the reporter recapitulates the endogenous expression of Vegfr3 in a subset of BECs, for example sinusoidal ECs (e.g. liver) or tip cells of angiogenic sprouts (e.g. developing retina ).

We thank the Reviewer for the suggestion. To analyze Vegfr3-tdTomato expression in these tissues, we performed wholemount staining of the explanted retina and on vibratome sections of the liver of transgenic mice. These data are included in the manuscript and we now show that the transgene was not detectable on the VEGFR-3-expressing BECs at the sprouting retinal front and only weakly labeled liver sinusoidal endothelial cells. It is therefore an excellent marker for LECs with little interference by expression on BECs, i.e. the Vegfr3-tdTomato transgenic mouse line is a valuable tool to study specifically lymphatic vessels.

2. In Fig 3, it seems that some VEGFR3+LYVE1+ LECs are not positive for tdTom (or anti-RFP staining in this case). This is particularly evident in Fig 3C but can also be seen in Fig 3A. Overall, RFP signal intensity seems to be heterogenous and not necessarily correlating with VEGFR3 staining intensity. Could the authors comment on this? Flow cytometric analysis of tdTom+ LECs could be useful .

We thank reviewer #1 for raising this important issue. As suggested, we analyzed single cell preparations from lung by cytometry. Indeed, in a CD31pos, PodoplaninHigh gate that should only contain double positive bona fide LECs, still more than half of the cells did not score positive for tdTomato, based on the endogenous fluorescence of the marker protein. We presume that a fraction of the cells was genuinely not labelled because insufficient promotor activity, while some of the cells may have been missed due to insufficient sensitivity of the flow cytometer. Clearly the proportion of positive cells was higher after immune staining. We nevertheless emphasize that the reporter faithfully labels all detectable lymphatic vessels in various tissues, hence provides some beneficial lymph vessel specificity.

Minor comments:

1. line 244: “… the epicardium but hardly invade into the myocardium are described for larger mammals.” I believe it is a typo and should be “… the epicardium but hardly invade into the myocardium as described for larger mammals.

We thank the reviewer for carefully proof-reading the manuscript, the mistake has been corrected.

Reviewer #2: 

Redder et al report generation of Vegfr3-tdTomato transgenic mice and demonstrate utility of such model for visualization of lymphatic vessels in different organs using a variety of imaging approaches, such as confocal, light sheet and two-photon microscopy. Overall, the results are clearly presented and the manuscript is well written. The model will be a useful addition to the arsenal of tools for characterization of lymphatic vessels, especially in adult animals and pathological conditions. I have only few comments as detailed below.

1. It seems that a proportion of LECs does not express the reporter (e.g. Figure 3 C), could the authors quantify the proportion of such tdTomato-negative LECs using FACS?

Quantification by flow cytometry has been performed, please see our response to reviewer #1.

2. In addition to LECs, Vegfr3 expression has been reported in the tips of sprouting blood vessels in postnatal retina and tumors, in some fenestrated blood endothelial cells, immune cells and neural stem cells in subventricular zone of brain. Do the authors observe such pattern of expression in their model?

We have probed expression in the tips of sprouting retinal blood vessels and liver sinusoidal endothelium, these data have now been included in the manuscript and provide string evidence for a lymph vessel specific activation of the reporter gene.

3. The authors should discuss in more detail advantages and disadvantages of their model in comparison to other existing lymphatic reporter strains. For example, expression of Prox1 in multiple non-endothelial cell types sometimes precludes use of Prox1- based reporters.

We thank reviewer #2 for this suggestion. We have now discussed these aspects in detail in lines 325-338 and 343-355 of the discussion, which put our Vegfr3-tdTomato mouse in perspective to other published reporter lines, especially those based on Prox1 promoter expression.

---

## [Decision Letter · Decision Letter 1]

6 Sep 2021

Vegfr3-tdTomato, a reporter mouse for microscopic visualization of lymphatic vessel by multiple modalities

PONE-D-21-08111R1

Dear Dr. Kiefer,

I apologize for the delayed response, but we’re pleased to inform you that your manuscript has been judged scientifically suitable for publication and will be formally accepted for publication once it meets all outstanding technical requirements.

Kind regards,

Stefan Liebner, PhD

Academic Editor

PLOS ONE

Additional Editor Comments (optional):

Reviewers' comments:

Reviewer's Responses to Questions

**Comments to the Author**

1. If the authors have adequately addressed your comments raised in a previous round of review and you feel that this manuscript is now acceptable for publication, you may indicate that here to bypass the “Comments to the Author” section, enter your conflict of interest statement in the “Confidential to Editor” section, and submit your "Accept" recommendation.

Reviewer #1: All comments have been addressed

2. Is the manuscript technically sound, and do the data support the conclusions?

Reviewer #1: Yes

3. Has the statistical analysis been performed appropriately and rigorously? 

Reviewer #1: N/A

4. Have the authors made all data underlying the findings in their manuscript fully available?

Reviewer #1: Yes

5. Is the manuscript presented in an intelligible fashion and written in standard English?

Reviewer #1: Yes

6. Review Comments to the Author

Reviewer #1: (No Response)

7. PLOS authors have the option to publish the peer review history of their article (what does this mean?). If published, this will include your full peer review and any attached files.

Reviewer #1: No

---

## [Editor Report · Acceptance letter]

10 Sep 2021

PONE-D-21-08111R1 

Vegfr3-tdTomato, a reporter mouse for microscopic visualization of lymphatic vessel by multiple modalities 

Dear Dr. Kiefer:

I'm pleased to inform you that your manuscript has been deemed suitable for publication in PLOS ONE. Congratulations! Your manuscript is now with our production department. 

Kind regards, 

on behalf of

Dr. Stefan Liebner 

Academic Editor

PLOS ONE